# A High-Efficiency, Ultrawide-Dynamic-Range Radio Frequency Energy Harvester Using Adaptive Reconfigurable Technique

Qian Lian [1,2] and Niansong Mei [1,2,*]

1 Shanghai Advanced Research Institute, Chinese Academy of Sciences, Shanghai 201210, China
2 University of Chinese Academy of Sciences, Beijing 100049, China
* Correspondence: meins@sari.ac.cn

**Abstract:** This paper presents a novel adaptive reconfigurable rectifier architecture for radio frequency energy harvesting (RFEH); in addition, a new metric for high-efficiency dynamic range (DR) is proposed. The presented rectifier architecture is based on a double-sided diode-feedback cross-coupled differential-drive rectifier (CCDR) structure incorporating self-body bias for reconfigurable operation. An adaptive structure based on a Schmitt trigger is proposed to adaptively switch the rectifier connection without auxiliary voltage ($V_{aux}$), with two rectifier stages in parallel at low power and in series at high power. The system is simulated at a 180 nm CMOS process and the results show more than 17 dB DR at 900 MHz, with efficiency higher than 50% at a 100 kΩ load.

**Keywords:** RFEH; rectifier; adaptive controller; wide dynamic range

## 1. Introduction

The rapid development of the Internet of Things (IoT) has led to the massive deployment of IoT low-power nodes such as smart home equipment, wearable devices, and medical implantable products. The conventional way of powering these nodes using batteries has obvious drawbacks of unsustainability, environmental pollution, and volume occupation. Therefore, finding a more suitable way of powering these nodes has become an issue that needs to be solved urgently [1–6]. Passive IoT provides a solution for this which is able to harvest energy from the environment to feed low-power sensors in a wireless sensor network in order to achieve self-powered nodes without batteries [7–11]. RF energy is one of the accessible environmental energy sources and has the advantage of having been widely used in RFID, a case of passive IoT [12–15]. Secondly, it is ubiquitous in the real environment and can be harvested everywhere, which greatly broadens the application range of sensor nodes. However, due to the low energy density of RF energy itself, coupled with the energy attenuation and disparity of distribution caused by obstacles like mountains, buildings, etc., RFEH is faced with the problem of poor quality and inconsistency of the energy source, which puts high demands on the power conversion efficiency (PCE) and DR of the RFEH system. At the same time, these two points become the main challenges of RFEH [16,17].

The RFEH block diagram is shown in Figure 1. The purpose of the impedance matching network is to match the input impedance of the rectifier to the antenna impedance for maximum power transfer. The rectifier is the critical module in the RFEH system and the DC–DC converts the low output voltage of the rectifier into a usable voltage for the sensor. There are two types of basic structures for the rectifier: CCDR and Dickson rectifier. CCDRs were proposed in [18], the highest PCE can be up to 80%, but they have a small DR; the Dickson rectifier [19], compared to the CCDR, is more suitable for the acquisition of the high-input-power range, the low input power under the Dickson structure is not advantageous [20]. As there are complementary properties of the two structures, it has been the focus of many studies to build wide-range energy harvesting systems by combining

the characteristics of the two structures. A. Choo et al. utilized a structure combining the CCDR and Dickson topologies to improve the DR performance of the rectifier in [21,22], yet the disadvantage was that the design required $V_{aux}$, which greatly limited the application scenarios of the system, regardless of whether the $V_{aux}$ came from batteries or other ambient energy sources, thus leaving the system without passive operation. A. Choo et al., in [23], proposed another structure that adaptively selects the CCDR or Dickson structure at the last stage, achieving a DR of 21 dB at an efficiency above 20% and a 79.9% peak PCE without the need for an external power supply; however, this structure has a smaller DR at high efficiency. D. Khan et al. used two identical rectifiers for switching, with the same disadvantage of requiring $V_{aux}$ [24]. In addition to the strategy of combining the two structures to improve DR, M. H. Ouda et al. and A. S. Almansouri et al. proposed the use of CCDR-based resistor feedback [20] and diode feedback [25], respectively, to improve DR, which not only improves the efficiency at high power, but also possesses an excellent efficiency performance, similar to the CCDR at low power, with no external power supply involved. Furthermore, S. M. Noghabaei et al. used dynamic and static compensation techniques to reduce the transistor threshold voltage to improve the rectifier performance, but only achieved 42.4% efficiency at a 450 kΩ load [26]. In addition, the use of the body biasing technique also reduces the losses during RF–DC operation, and thus, improves the PCE and DR. In [27], Amin Khalili Moghaddam et al. describe the principle of body biasing in detail and use the outputs of the CCDR structure as the voltage of the body bias for inter-stage biasing to achieve the rectification efficiency improvement. The RF–DC circuit in [28] is also based on the CCDR structure for body biasing, but unlike in [27], the body bias voltage used in [28] is derived from the single-stage circuit itself. Yan Li et al. also use body biasing to achieve PCE improvement in [29], but with a narrower DR.

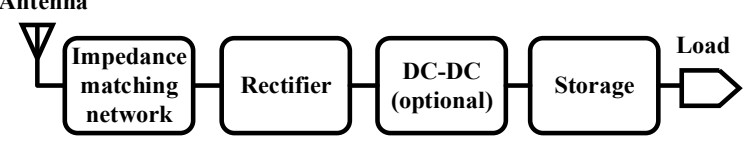

**Figure 1.** Block diagram of RFEH system energy conversion.

In this paper, an adaptive reconfigurable RFEH rectifier based on an improved CCDR structure is proposed, in which the fundamental module utilizes diode-feedback and self-body-biasing techniques to simultaneously improve the DR and PCE performance. In addition, the adaptive technique is utilized to automatically switch the rectifier connection to obtain wide DR under passive conditions. The principle of the proposed architecture is explained in Section 2, the simulation results are given in Section 3 for comparison with the literature, and the conclusions are presented in Section 4.

## 2. Proposed RF Energy Harvester Architecture

### 2.1. System Design

In order to broaden the application scenarios of the RF energy harvester so that it can receive a wide range of input power, and thus, increase the energy obtained by the load, the DR at high power is extended by using an architecture with two rectifiers in parallel at low power and in series at high power, taking advantage of the fact that the peak efficiency shifts to high power when two rectifiers are connected in series [24]. The purpose of the system is to obtain high DR at high efficiency, which requires better PCE and DR performance for both parallel and series paths. A diode-feedback CCDR structure incorporating the self-body-biasing technique is proposed as a single-stage rectifier to improve the PCE and DR at low input power. Finally, to get rid of $V_{aux}$ to enable the system to operate independently, an adaptive control circuit is proposed to automatically adjust the rectifier connection according to the output. Figure 2 illustrates the block diagram of the RF harvester in this paper, including the impedance matching network, two single-stage rectifiers, three transmission gate switches for switching paths, and an adaptive control

circuit, which enables adaptive switching between two states of two-stage rectifiers in series for high input power and two-stage rectifiers in parallel for low input power.

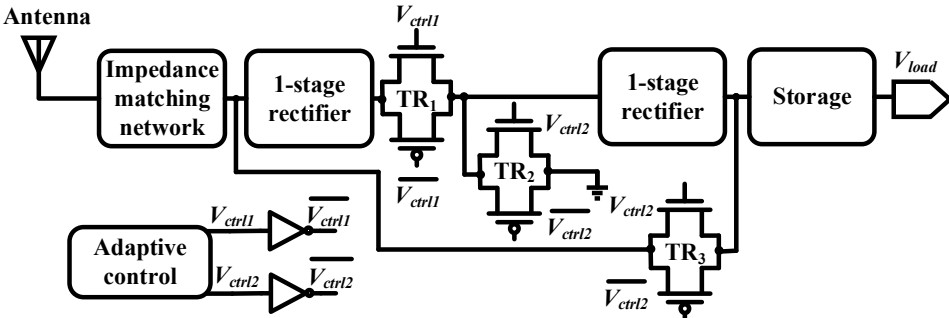

**Figure 2.** System block diagram.

### 2.2. Single-Stage Rectifier Structure

Figure 3 shows the conventional CCDR and diode-feedback CCDR. The CCDR structure is characterized by a high peak PCE, and it is shown in Ref. [18] that the peak PCE of the CCDR can reach greater than 80% at 953 MHz and 100 kΩ load, but the DR is very small. This characteristic of CCDR is due to the two cross-coupled pairs of MOSFET, used to reduce or increase the on-resistance according to the direction of the input voltage to increase the PCE, which will greatly improve the efficiency at low power inputs; however, at high power, this structure will cause a large leakage current, resulting in a reduction in the PCE at high power, and thus, the DR is not large. For example, when the positive-cycle $M_{P1}$ and $M_{N2}$ are on, and $M_{P2}$ and $M_{N1}$ are off, the negative voltage at RF− will reduce the on-resistance of $M_{P1}$ and increase the on-resistance of $M_{N1}$, and the positive voltage at RF+ will reduce the on-resistance of $M_{N2}$ and increase the on-resistance of $M_{P2}$. The final effect is to reduce the on-resistance of the positive conduction path and increase the on-resistance of the off-path, which results in a significant reduction in operating losses and an increase in PCE. However, when the input is high power, $V_{out,rect}$ rises, and when the difference between $V_{out,rect}$ and the $M_{P1}$ gate is greater than $|V_{thp1}|$ to turn $M_{P1}$ on, $M_{P1}$ reverses conduction to generate leakage current. This reverse leakage current in the CCDR structure greatly limits the circuit's PCE at high power inputs and limits the value of DR.

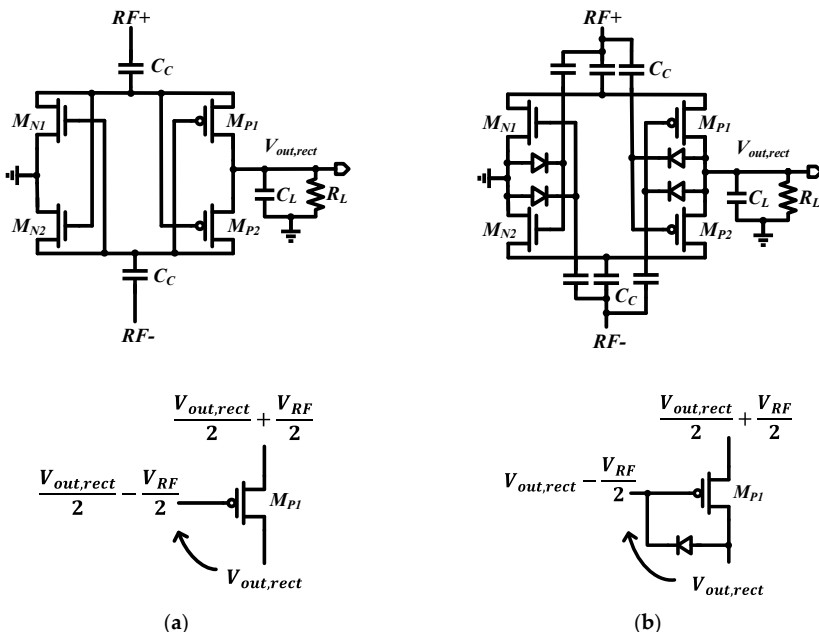

**Figure 3.** (**a**) Conventional CCDR and analysis [18]; (**b**) diode-feedback CCDR and analysis [25].

CCDR was compared with diode-feedback CCDR in [25], which showed that the diode-feedback structure can greatly extend the DR of the rectifier without significantly reducing the peak PCE. The main idea of diode feedback is to use the diode-bias voltage applied to the PMOS gate to reduce the leakage current at high power, and thus, increase the PCE at high power to improve the DR. When in low-power conditions, the diode cuts off and the rectifier operates in the conventional CCDR mode, which guarantees a higher PCE at low-power conditions. When the diode conducts at high power, as shown in Figure 3a, the voltage difference between the source and gate that determines the magnitude of the leakage current in the conventional structure is $V_{out,rect} - \left( \frac{V_{out,rect}}{2} - \frac{V_{RF}}{2} \right) = \frac{V_{out,rect}}{2} + \frac{V_{RF}}{2}$ in

$M_{P1}$, for example, while in Figure 3b this value is $V_{out,rect} - \left( V_{out,rect} - \frac{V_{RF}}{2} \right) = \frac{V_{RF}}{2}$ in the diode-feedback structure, which significantly reduces the $V_{SG}$ at $M_{P1}$ reverse conduction, making the conditions for the generation of leakage current harsher, and significantly reducing the power loss due to leakage current. The result is that there is no significant drop in PCE at high voltage and the circuit can maintain high efficiency over a wider DR range.

In order to improve the PCE at low power, the key is to reduce the loss in the current path, which can be achieved by adjusting the threshold voltage ($V_{th}$) of the MOSFET. This method is also known as the self-body-biasing technique [27–29]. The main principle is to take advantage of the source–body voltage difference of NMOS and PMOS to reduce $V_{th}$ when the MOSFET is on, and to increase $V_{th}$ when the MOSFET is off. The expressions for $V_{th}$ of NMOS and PMOS are shown in Equations (1) and (2).

$$V_{thn} = V_{th0n} + \gamma \left( \sqrt{|2\varnothing_F| + V_{sb}} - \sqrt{|2\varnothing_F|} \right) \tag{1}$$

$$V_{thp} = V_{th0p} - \gamma \left( \sqrt{|2\varnothing_F| - V_{sb}} - \sqrt{|2\varnothing_F|} \right) \tag{2}$$

In this paper, the self-body-biasing technique is utilized on the basis of the diode-feedback CCDR structure to improve the PCE of a single-stage RF–DC rectifier by superimposing bias voltages from the RF source at the bulk terminals of NMOS and PMOS, respectively. As illustrated in Figure 4, when the $M_{P1}$ and $M_{N2}$ conduct in the positive half-cycle, the negative voltage superimposed on the $M_{P1}$ bulk terminal makes the voltage difference between $M_{P1}$ source and body ($V_{sbp1}$) increase, and according to Equation (2), it is derived that $V_{thp1}$ increases, that is, $|V_{thp1}|$ decreases, which is favorable for the $M_{P1}$ to conduct, and the positive voltage superimposed on the $M_{N2}$ bulk terminal makes $V_{sbn2}$ decrease, and then $V_{thn2}$ decreases, which reduces the resistance of the conduction path in general, and improves the PCE. It is worth noting that the parasitic diode between the PMOS source and bulk, shown by the dashed line in Figure 4, precisely serves to balance the DC component of $V_{sb}$, so that the value of $V_{sb}$ is only affected by the RF AC component, which better assists the body effect in regulating $V_{th}$. In addition, this method of increasing the PCE, and thus, DR of the rectifier circuit also does not require any external assistance, as the bias voltage applied to the body terminal is derived from the RF input, so the circuit achieves an increase in its PCE and DR without consuming external energy. The circuit size is shown in Figure 4, the size of the MOSFET is derived from the iterative method, $C_C$ is used as a flying capacitor, the value is generally around 500 fF, the role of $C_B$ is to couple the energy from the RF side to the body of the MOSFET, simulation results show that a value of 1 pF is sufficient to transmit RF energy. The size of the diode affects its threshold voltage; a large threshold voltage will make the proposed diode-feedback with self-body-biasing circuit more likely to behave as a CCDR structure, and iteration yields that the rectifier output voltage is maximized when the diode size is chosen to be the smallest.

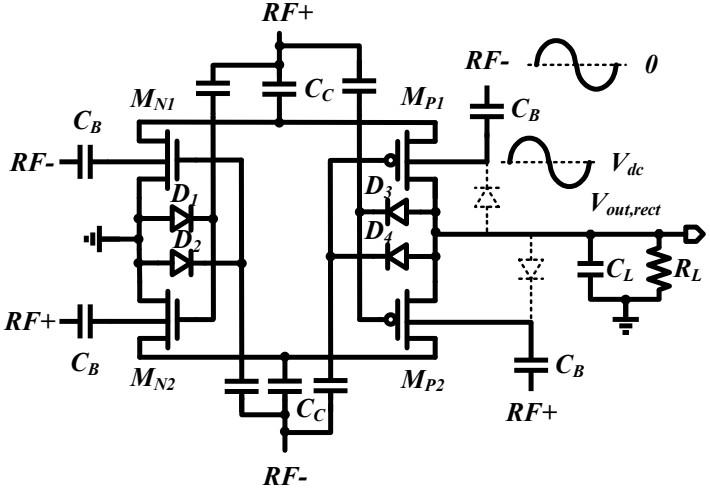

| $M_{N1,2}$ | 1μm/180nm |
|---|---|
| $M_{P1,2}$ | 2μm/180nm |
| $C_C$ | 500fF |
| $C_B$ | 1pF |
| $D_{1-4}$ | 485nm/420nm |

**Figure 4.** Diode-feedback rectifier structure with added self-body-biasing technique.

In order to evaluate the performance of the proposed diode-feedback with self-body-biasing structure more comprehensively, Figure 5 illustrates the graphs comparing the efficiency of the structure with the diode feedback under 100 kΩ, 200 kΩ, and 50 kΩ loads. From Figure 5, it can be seen that the efficiency of the proposed diode-feedback with self-body-biasing structure is improved at all three loads. At a load of 100 kΩ, the peak PCE moves from the original −17 dBm to −19 dBm, which is 2 dB in the direction of low power, and the overall efficiency is generally improved by 5% to 10% under low-power conditions, while there is no degradation of the efficiency at high power, which significantly improves the DR performance. With a load of 200 kΩ, the peak PCE moves from the original −18 dB to −20 dB, also moving 2 dB in the low power direction, with an overall efficiency improvement of about 5% at low power, which also improves the DR performance. At a load of 50 kΩ, the peak efficiency shifts from −14 dB to −16 dB, by 2 dB in the low-power direction, and the PCE improves by about 5% at low power and slightly at high power. Comparison of the efficiency curves at the three loads shows that the proposed diode-feedback with self-body-biasing structure significantly improves the DR performance of the circuit and the DR improvement is more obvious at low power. When the system is in the state of low-power two-stage rectifiers in parallel, this structure can have the effect of promoting DR at low power, which is exactly what we want.

### 2.3. Adaptive Controller

The adaptive control circuit serves to automatically switch the connection of the rectifier. It consists of two modules, a proposed adaptive structure based on a Schmitt trigger and a non-overlap structure that prevents overlapping switching conduction.

The adaptive control of this system needs to achieve the function that when the system output voltage reaches the switching voltage of the rectifier in parallel and series, there should be a signal that jumps from 0 to 1 to supply the transmission gate to realize the switching of the path. The Schmitt trigger can realize the function of detecting the input voltage and output jumping when the threshold is reached. Figure 6 shows the proposed

adaptive structure based on the Schmitt trigger, and a Schmitt trigger structure is shown on the right-hand side. The output of the Schmitt trigger is conditioned to jump from 0 to 1 by $M_2$ conduction, and the jump threshold is determined by the ratio of the sizes of $M_1$ to $M_5$ [30]. In this system, the high-level voltage at the output of the trigger should be the same as the voltage at the output of the system, so $V_{dd}$ of the Schmitt trigger should be charged by $V_{load}$. In this case, the relationship between $V_{in,schmitt}$ at the output of the Schmitt trigger jump and its decision parameter is as shown in Equation (3).

$$\frac{W_1 L_5}{L_1 W_5} = \left( \frac{V_{in,schmitt}}{V_{load} - V_{in,schmitt} - |V_{th1}|} \right)^2 \tag{3}$$

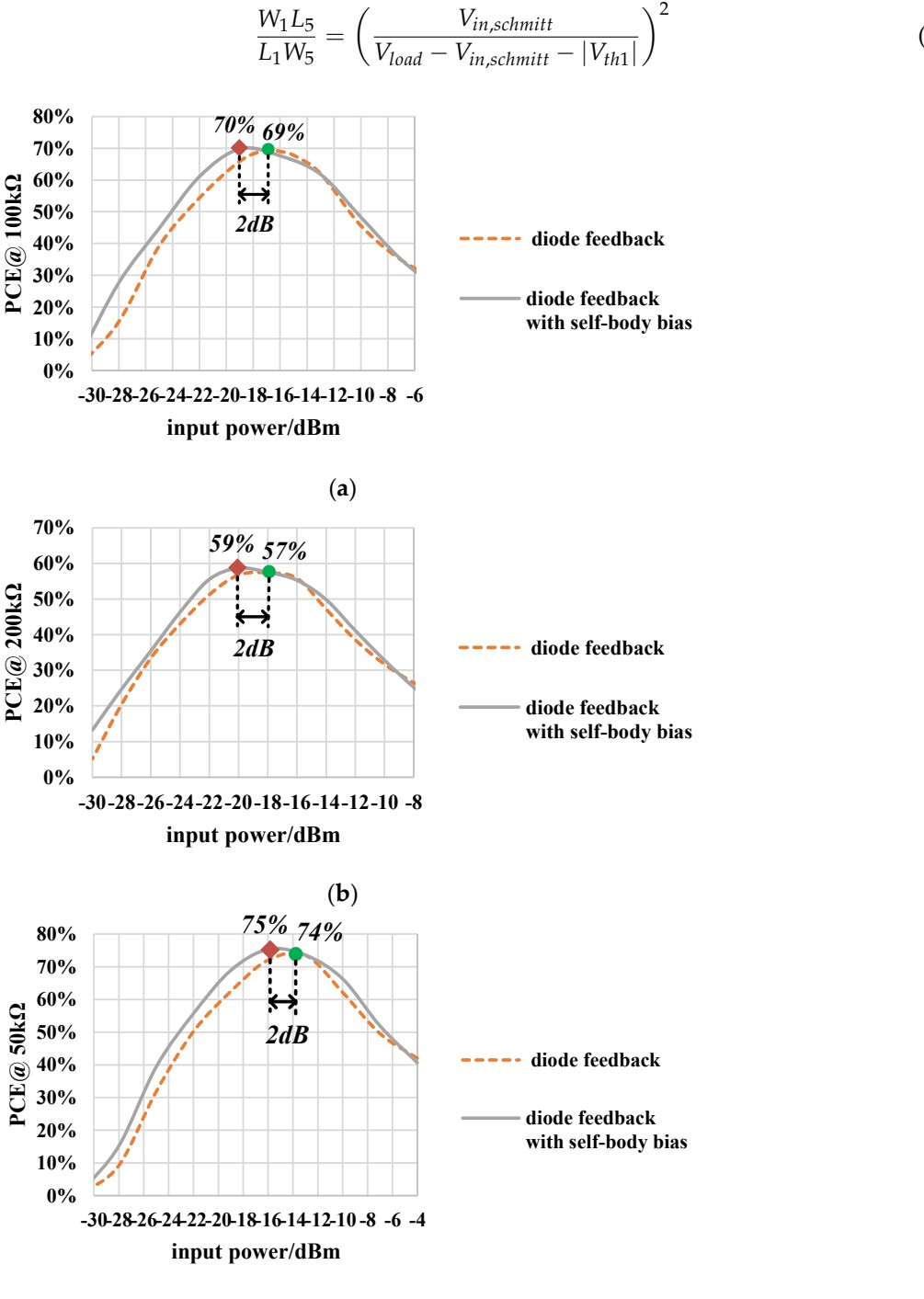

(a)

(b)

(c)

**Figure 5.** Comparison of diode-feedback and diode-feedback with self-body-biasing structures: (**a**) 100 kΩ load, (**b**) 200 kΩ load, (**c**) 50 kΩ load.

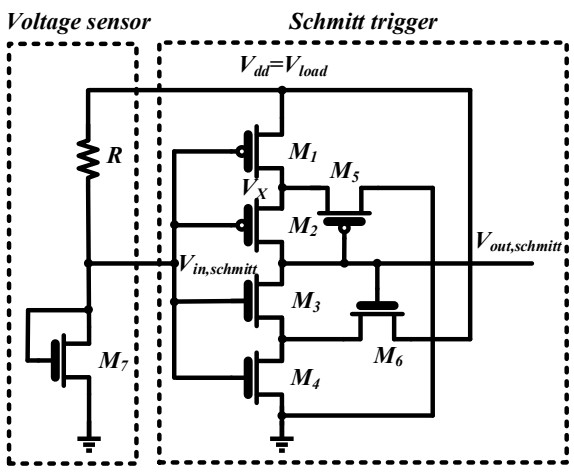

| M$_{1,2}$ | 10µm/300nm |
|---|---|
| M$_{3,4}$ | 5µm/350nm |
| M$_5$ | 220nm/300nm |
| M$_6$ | 2µm/350nm |
| M$_7$ | 40µm/350nm |
| R | 2MΩ |

**Figure 6.** Proposed adaptive structure based on Schmitt trigger.

It can be seen that to determine the value of $V_{in,schmitt}$, the relationship between $V_{load}$ and $V_{in,schmitt}$ needs to be established. In addition, the signal to be detected in this system is $V_{load}$, whereas in the original structure of the Schmitt trigger, the signal to be detected is the input, but it is not possible to connect $V_{load}$ directly to $V_{in,schmitt}$, which would prevent $M_2$ from conducting and block the path to the output pull-up. As a result, based on the above two perspectives, providing another pathway to correlate $V_{load}$ with $V_{in,schmitt}$ becomes an indispensable design to realize the function of the system. The left voltage sensor part in Figure 6 provides the relationship between $V_{load}$ and $V_{in,schmitt}$ in different cases, which consists of resistor R and the diode connection of $M_7$.

The objective of the system is that the output does not jump when $V_{load}$ is low and the output jumps when $V_{load}$ is high. Equation (3) defines the threshold value of $V_{in,schmitt}$, and the moment it is established occurs at the instant when $M_2$ is about to turn on, so the method of determining the relationship between $V_{load}$ and $V_{in,schmitt}$ when the output jumps is feasible and intuitive by measuring the conditions under which $M_2$ conducts. The output is 0 when $M_2$ does not conduct, and the output jumps to 1 when $M_2$ conducts. The condition for $M_2$ to turn on is shown in Equation (4).

$$V_X - V_{in,schmitt} = |V_{th2}| \qquad (4)$$

Since $V_X$ is less than $V_{load}$, $M_2$ must not conduct when $V_{in,schmitt}$ rises immediately after $V_{load}$, thus preventing $V_{out,schmitt}$ from rising, which is realized by the sampling resistor R in the voltage sensor. The relationship between $V_{load}$ and $V_{in,schmitt}$ is $V_{in,schmitt} = V_{load} - IR$, when $V_{load}$ is low, I changes slightly and $V_{in,schmitt}$ rises immediately after $V_{load}$. At this time, $V_{out,schmitt}$ will not jump, instead, when $V_{in,schmitt}$ rises with $V_{load}$ and exceeds the thresholds of $M_3$ and $M_4$, $V_{out,schmitt}$ will be pulled to ground, realizing the function of a $V_{out,schmitt}$ level of 0 under low power. The final level of the trigger output is 0, as shown in Figure 7, when the input is −20 dBm.

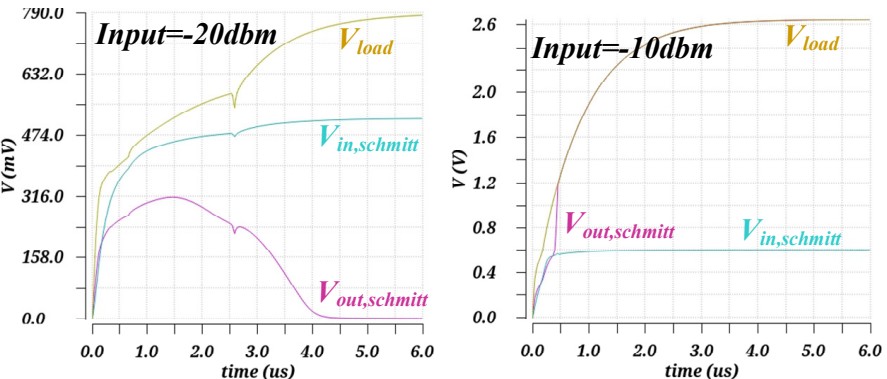

**Figure 7.** Waveforms of the proposed adaptive structure at low and high input powers.

To reduce the system power consumption, $M_7$ is set to be in the subthreshold region even when $V_{load}$ reaches its highest value. When $V_{load}$ and $V_{in,schmitt}$ rise to the point that $M_7$ enters the subthreshold region, the state of $M_7$ is $I_{D7} = I_{D0} \frac{W}{L} e^{\frac{V_{gs7}}{\xi V_T}}$. Due to the current limitation, the voltage across $M_7$ will be maintained at a stable value. $V_{in,schmitt}$ no longer rises with $V_{load}$, but is fixed by the voltage at the terminals of $M_7$, which gives $M_2$ a chance to conduct, and thus, make the output rise. When $V_{load}$ is higher than a certain value of $V_{in,schmitt}$, so that Equation (4) is established, $V_{out,schmitt}$ jumps, realizing the function of a $V_{out,schmitt}$ level of 1 under high power. As shown in Figure 7, when the input is $-10$ dBm, the jump occurs at 0.5 µs, and after 0.5 µs the trigger output is the same as $V_{load}$ with a level of 1.

The proposed adaptive structure based on a Schmitt trigger can automatically detect the value of $V_{load}$ to realize that the output is 0 for low voltage and 1 for high voltage. In this system, this trigger threshold is set to 1.2 V, which is the output voltage when switching between the series and parallel structures, and can be determined by adjusting the size of the transistors and resistor. This paper focuses on the procedure of the trigger output jumping from 0 to 1, the jump threshold is mainly determined by $M_{1,2,5}$, $M_7$, and R. The sizes of $M_3$ and $M_4$ affect the minimum input power of the system to start switching between series and parallel, larger $M_3$ and $M_4$ sizes make it easier for the input power to pass through, but $M_{1-4}$ should not be set too large to ensure that the path from $V_{dd}$ to ground has a certain value of resistance without creating too much leakage. The $M_6$ sizing affects the drop threshold of the trigger, which does not need to be considered in this paper. After iteration, the dimensionality diagram of the proposed adaptive structure based on a Schmitt trigger is shown in Figure 6. The adaptive triggering of this module allows the system to operate autonomously, free from external power sources.

As shown in Figure 8, in order to prevent current leakage due to simultaneous conduction of different switches at the switching moment, non-overlap structure is used to control $TR_1$, $TR_2$, and $TR_3$ in parallel and series paths, so that $TR_1$ is switched individually and $TR_2$ and $TR_3$ are switched at the same time, which reduces the current loss at the switching moment.

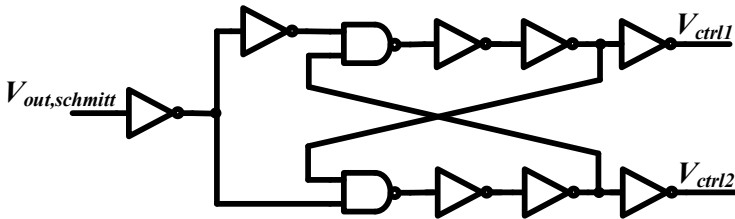

**Figure 8.** Non-overlap control [31].

### 3. Simulation Results

The system operates at 900 MHz and a comparison of the simulation curves of the proposed switching efficiency with those of the parallel and series circuits is shown schematically in Figure 9, from which it can be seen that the system achieves the ideal switching effect. Due to the presence of control circuit losses, the switching power point is shifted to the right to become −15 dBm, where the efficiency at the lowest point is 51%. In addition, the proposed system shows superior DR performance. The system efficiency reaches 50% at the −23 dBm input, so the power supplied to the output at this point is 2.5 μW, which is four times the power that can be supplied at 20% efficiency in [23], and it is enough for some low-power nodes to operate. For the loads to receive high enough power even at low-power conditions, the system viability is, therefore, better reflected by the use of PCE over 50% as an indicator of system DR.

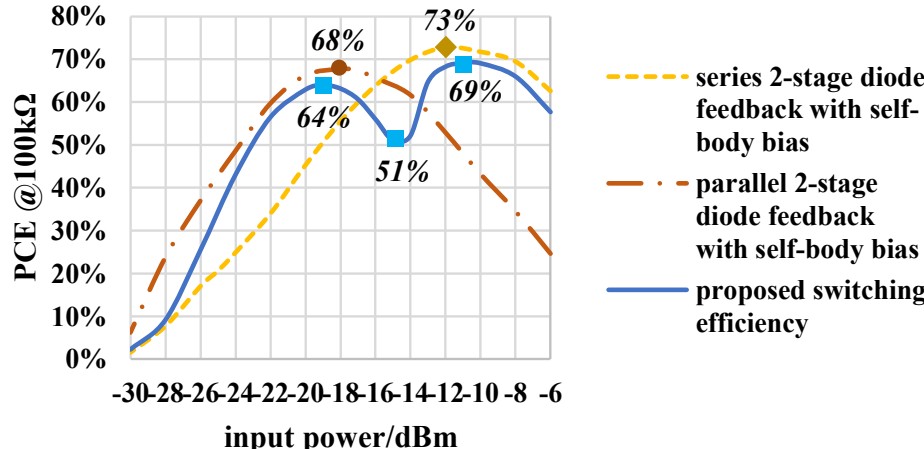

**Figure 9.** Efficiency comparison of the proposed switching structure with series and parallel structures.

Because of the threshold limitations of $M_3$ and $M_4$ in the adaptive Schmitt trigger, the system starts switching between series and parallel structures normally from −26 dBm input: the two rectifiers are connected in parallel in the case of −26 dBm to −15 dBm input; and the two rectifiers are connected in series in the case of the input power being greater than −15 dBm. The proposed rectifier exhibits dual PCE peaks, reaching 64% PCE at −19 dBm and 69% PCE at −11 dBm under 100 kΩ load at 900 MHz frequency. Meanwhile, the efficiency is above 50% from −23 dBm to greater than −6 dBm, achieving an ultra-wide DR of more than 17 dB. It is important to note that unlike the work in [21,22,24] the excellent performance achieved by the proposed system is built on the basis that $V_{aux}$ is not required, which greatly broadens the application scenarios of the system. Table 1 lists a comparison between the proposed system and the recently released CMOS RFEH rectifiers operating in the GSM900 band. In terms of DR, the RF energy harvester proposed has a superior bandwidth to all those proposed in the literature without the need of $V_{aux}$. The DR listed from the comparative literature are all estimated from the figure. Since this paper focuses on the improvement of DR performance and pays more attention to the expansion of the system's application range, the requirement of PCE is not strict. The single-stage structure is then dedicated to DR improvement, and the proposed adaptive structure uses two-stage switching technique to extend the DR of the system as well; therefore, the peak PCE of the system is not really high, and it can also be seen in Table 1 that the peak PCE is lower than those reported in the literature [22,23,32,33], which is a trade-off. Nevertheless, the double-peak PCE of more than 60% in this paper also shows some competitiveness.

**Table 1.** Performance comparison with previously published work.

| | CMOS Technology | Frequency | Topology | No. of Stages | Output Load | $V_{aux}$ | Peak PCE | DR ($\geq$50%) |
|---|---|---|---|---|---|---|---|---|
| This Work ** | 180 nm | 900 M | Self-Biasing, Double Sided, Reconfigurable | 1–2 | 100 kΩ | No | 64%@−19 dBm<br>69%@−11 dBm | (−23~>−6) >17 |
| MTT'18 [25] * | 180 nm | 900 M | Self-Biasing, Double Sided | 1 | 100 kΩ | No | 66%@−18.5 dBm | (−22.6~−15) 7.6 |
| VLSI'22 [21] * | 130 nm | 900 M | Dickson, Reconfigurable | 6–12 | 100 kΩ | Yes | 34.93%@−10 dBm | -- |
| TCAS-I'22 [26] * | 130 nm | 915 M | Cross-Coupled Voltage Compensation | 10 | 450 kΩ | No | 42.4%@−16 dBm | -- |
| VLSI'23 [22] | 130 nm | 900 M | Cross-Coupled Dual Topology | 3 | 100 kΩ | Yes | 78.4%@−16 dBm *<br>88%@−16.5 dBm ** | (−19~−13.3) 5.7 *<br>(−19.5~−7.3) 12.2 ** |
| TCAS-II'23 [23] * | 65 nm | 900 M | Cross-Coupled Advanced Topology Amalgamation | 3 | 100 kΩ | No | 79.77%@−17.5 dBm | (−22~−11.7) 10.3 |
| AICSP'23 [32] ** | 180 nm | 920 M | Body Control | 1–3 | 100 kΩ | No | 71.2%@−15.6 dBm | (−18~−12) 6 |
| AICSP'24 [33] ** | 180 nm | 900 M | Diode-Feedback and Feed-Forward Rectifier | 1 | 100 kΩ | No | 76.13%@−19 dBm | (−24~−14) 10 |

\* Measurement result; ** simulation result.

## 4. Conclusions

In this paper, a novel adaptive reconfigurable CMOS rectifier is presented, in addition, a new DR metric capable of providing sufficient power to the load is proposed. The proposed RF energy harvester does not require an external power supply, $V_{aux}$, to realize the DR improvement. The DR of the single-stage rectifier is improved by using a diode-feedback structure incorporating self-body bias as the base structure. The effect of switching the rectifier connection mode while eliminating $V_{aux}$ is achieved by utilizing the proposed adaptive technique based on a Schmitt trigger structure. Simulation results show that the first time the PCE exceeds 50% is at −23 dBm. A peak PCE of 64% is achieved at −19 dBm, with the rectifiers connected in parallel at low power, and a 69% peak PCE is attained at −11 dBm, with the rectifiers connected in series at high power. An efficiency of 58% is still achievable up to −6 dBm, demonstrating the superior performance of this system. In order to extend this paper, future work could configure the rectifier to other commonly used frequency bands so that the RF–DC circuits can harvest energy from multiple frequency bands to extend the range of the rectifier's applications. In addition, the current multistage switching circuits for harvesting are limited to a fixed detection voltage to switch the number of stages; to make RF energy harvesting more practical, an adaptive harvesting network for different loads is a promising direction.

**Author Contributions:** Conceptualization, Q.L. and N.M.; methodology, Q.L. and N.M.; software, Q.L.; validation, Q.L.; formal analysis, Q.L.; investigation, Q.L.; resources, Q.L.; data curation, Q.L.; writing—original draft preparation, Q.L.; writing—review and editing, Q.L. and N.M.; visualization, Q.L.; supervision, N.M.; project administration, N.M. All authors have read and agreed to the published version of the manuscript.

**Funding:** This research received no external funding.

**Data Availability Statement:** Data are contained within the article.

**Conflicts of Interest:** The authors declare no conflicts of interest.

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
