# Peer review of "A High-Efficiency, Ultrawide-Dynamic-Range Radio Frequency Energy Harvester Using Adaptive Reconfigurable Technique"

_electronics, doi:10.3390/electronics13071193_

Round 1

Reviewer 1 Report

Comments and Suggestions for Authors

The paper provides a comprehensive overview of the challenges and solutions in RF energy harvesting (RFEH) systems for powering low-power IoT nodes. The incorporation of diode-feedback and self-body-biasing techniques adds novelty to the proposed architecture, potentially offering significant improvements over conventional rectifier designs. However, the reviewer has some comments and suggestions for the authors to improve the quality and technicality of the manuscript.

·        Although the comparison between different rectifier structures and techniques for improving DR and efficiency is insightful, it would be beneficial to include a more quantitative comparison, if possible, such as efficiency values and dynamic range improvements achieved by each approach, to provide a clearer understanding of their performance.

·        While the proposed architecture sounds promising in theory, the practical implementation details are not thoroughly discussed. It would be beneficial for the authors to provide insights into the feasibility and practical challenges of implementing the adaptive reconfigurable rectifier system.

·        The paper mentions a comparison between the diode-feedback structure and the diode-feedback self-body-biasing structure, showing improved efficiency under low-power conditions. However, the analysis seems limited to efficiency curves at a single load impedance of 100kΩ. A more comprehensive performance analysis across a range of load impedances and input power levels would provide a more complete understanding of the proposed technique's effectiveness.

·        The comparison with the literature should include not only performance metrics but also a discussion of the advantages and limitations of the proposed approach compared to existing methods. For instance, are there any trade-offs in terms of complexity, manufacturing feasibility, or sensitivity to process variations? Acknowledging these limitations would contribute to a more balanced assessment of the proposed approach.

·        The conclusion should not only summarize the findings but also offer insights into potential future research directions. What are the next steps in refining the proposed technique or extending its application to different scenarios or technologies? Providing suggestions for future work would enrich the paper's contribution to the field.

Comments on the Quality of English Language

The paper should be revised for grammatical errors and punctuation. 

Reviewer 2 Report

Comments and Suggestions for Authors

The original research paper deals with radio frequency (RF) energy harvesters. An improved rectifier architecture for RF energy harvesters is proposed and verified by using simulations. All the references are appropriate and cited correctly in the manuscript. Additionally, there are no any excessive self-citations. English language has some minor mistakes.

In order to improve quality of the manuscript, the authors should follow the comments below:

1) Only simulation results are presented in the manuscript. In order for the paper to be more interesting, useful and plausible it is necessary to make experimental verification of the simulation results.

2) Performance comparison with previously published works is done in Table 1. It is not a good idea to compare simulation results obtained in the paper with the experimental results obtained in other research papers. It is much better to make experiments and then compare the experimental results with that of other research papers.

3) Reference list should be formatted according to MDPI style.

4) Figures 3 and 8 have schematic diagrams copied from other papers. Do the authors of the manuscript have permission to copy the images from the copyright holders?

Comments on the Quality of English Language

Minor editing of English language required.

Round 2

Reviewer 1 Report

Comments and Suggestions for Authors

Thanks to the authors for their thorough response to the reviewer's comments. All my comments are addressed. I have no more comments. The paper is improved by addressing all the comments and suggestions. The reviewer thinks that the paper is ready to be accepted for publication. 

Author Response

Thank you very much for your review work and for recognizing.

The changes made this time are the explanation of the nomenclature about RF has been added in the abstract based on the comment from academic editor and the format of reference [33] has been modified .

Reviewer 2 Report

Comments and Suggestions for Authors

The authors responded satisfactorily on all of my original comments and now I can accept the paper. 

Author Response

(The authors gave the same response as above.)
